# Duplex Surface Modification of 304-L SS Substrates by an Electron-Beam Treatment and Subsequent Deposition of Diamond-like Carbon Coatings

**Stanislava Rabadzhiyska [1], Georgi Kotlarski [1], Maria Shipochka [2], Peter Rafailov [3], Maria Ormanova [1], Velichka Strijkova [4], Nina Dimcheva [5] and Stefan Valkov [1,6,\***

1. Institute of Electronics, Bulgarian Academy of Sciences, 72 Tzarigradsko Chaussee Blvd, 1784 Sofia, Bulgaria; s1983@abv.bg (S.R.); gvkotlarski@gmail.com (G.K.); m.ormanova@ie.bas.bg (M.O.)
2. Institute of General and Inorganic Chemistry, Bulgarian Academy of Sciences, Acad. Georgi Bonchev Str., bld. 11., 1113 Sofia, Bulgaria; shipochka@svr.igic.bas.bg
3. Institute of Solid State Physics, Bulgarian Academy of Sciences, 72 Tsarigradsko Chaussee Blvd, 1784 Sofia, Bulgaria; rafailov@issp.bas.bg
4. Institute of Optical Materials and Technologies, Bulgarian Academy of Sciences, 109 Acad. G. Bonchev Str., 1113 Sofia, Bulgaria; vily_strij@abv.bg
5. Department of Physical Chemistry, Plovdiv University Paisii Hilendarski, 24 Tzar Asen Str., 4000 Plovdiv, Bulgaria; ninadimcheva@gmail.com
6. Department of Mathematics, Informatics and Natural Sciences, Technical University of Gabrovo, 4 H. Dimitar Str., 5300 Gabrovo, Bulgaria
* Correspondence: stsvalkov@gmail.com

**Abstract:** In this study, we present the results of the effect of duplex surface modification of 304-L stainless steel substrates by an electron-beam treatment (EBT) and subsequent deposition of diamond-like carbon coatings on the surface roughness and corrosion behavior. During the EBT process, the beam power was varied from 1000 to 1500 W. The successful deposition of the DLC coatings was confirmed by FTIR and Raman spectroscopy experiments. The results showed a presence of C–O, C=N, graphite-like $sp^2$, and mixed $sp^2$-$sp^3$ C–C bond vibrations. The surface topography was studied by atomic force microscopy. The rise in the beam power leads to a decrease in the surface roughness of the deposited DLC coatings. The studies on the corrosion resistance of the samples have been performed using three electrochemical techniques: open circuit potential (OCP), cyclic voltammetry (polarization measurements), and non-destructive electrochemical impedance spectroscopy (EIS). The measured corrosion potentials suggest that these samples are corrosion-resistant even in a medium, containing corrosive agents such as chloride ions. It can be concluded that the most corrosion-resistant specimen is DLC coating deposited on electron-beam-treated 304-L SS substrate by a beam power of 1500 W.

**Keywords:** diamond-like carbon coating; surface modification; electron-beam treatment; electron-beam physical vapor deposition; surface topography; corrosion properties

## 1. Introduction

Metallic materials have been commonly used in biomedicine for a century owing to their desirable properties, which include high strength and toughness, fatigue resistance, and inertness [1,2]. Stainless steels, in particular 304-L, are very promising in modern biomedical engineering due to their biocompatibility, malleability, high heat stability, and excellent resistance to corrosion [3]. Because of these properties, 304-L stainless steel is widely used for producing dental and orthopedic implants as well as for specific biomedical applications. Despite its good properties, this material exhibits poor tribological properties and weak chemical bonds with human bones [4]. These drawbacks appear mostly on the surface of the implant materials, which can be overcome by an appropriate technique for

surface treatment. Surface modifications of biomaterials play a vital role in matching the complexities of the biological system and enhancing the performance of bio-implants.

A lot of methods for surface modification of materials exist. Currently, surface modification by high energy fluxes, such as electron, ion, and laser beams are widely used [5]. Electron-beam surface treatment technology has developed rapidly in recent decades due to its advantages over conventional methods, such as high efficiency and accuracy, very small size of the zones treated, high density of the input energy, high heating and cooling rates, good reproducibility and hardly any chemical pollution [6–8]. The author of [9] investigated the possibility of hardening carbon steels by electron-beam surface modification, and a numerical model was developed. The results showed that the heating and cooling rates depend weakly on the electron-beam power and are strongly influenced by the treated sample speed of motion. Higher values of the speed of motion of the specimen lead to increased cooling rate and microhardness, respectively. Proskurovsky et al. [10] investigated technology for the surface modification of metallic materials based on the use of low-energy, high-current pulsed electron beams. The results showed that this technique enhanced the strength and electrochemical properties of the modified materials.

Another method for surface modification of materials is the deposition of coatings on their surfaces [11]. The authors of [12] synthesized carbon nanostructures on H18 steel by the chemical vapor deposition (CVD) method. The results showed that the electrochemical behavior of the carbon structures strongly depends on the technological conditions, where the best corrosion properties were obtained at 700 °C.

Diamond-like carbon (DLC) films possess unique characteristics such as corrosion resistance, low friction coefficient, chemical inertness, and excellent smoothness [13]. Additionally, DLC coatings demonstrate high biocompatibility, which makes them suitable for implementation in orthopedic and dental medicine. According to the literature, DLC coatings [14] are hard and amorphous-like structures with a mixture of $sp^2/sp^3$ and hydrogen, where the amount of $sp^3$ bonds is significant. It is well known that the $sp^2$ hybridization represents one $s$ orbital and two $p$ orbitals, creating three new hybrid $sp^2$ orbitals with equal energy, and correspond to a graphitic structure. Similarly, in $sp^3$ hybridization, one s and three p orbitals form four new hybrid $sp^3$ orbitals and correspond to a diamond structure [15].

Recently, physical vapor deposition (PVD) technologies have been one of the main methods for the formation of coatings for biomedical applications. The authors in [16] deposited DLC films by plasma immersion and investigated the corrosion behavior and mechanical properties of the obtained coatings. The results indicated good resistance to corrosion and good tissue response of the DLC coatings proving the high biocompatibility of these materials. Similarly, the authors of [17] applied diamond-like carbon films on an Si substrate with different silver contents by radio frequency magnetron sputtering. It was proved that the presence of silver in the DLC films increased the conductivity and antimicrobial characteristics of the deposited coatings, but the hardness was reduced significantly. In [4], the PVD deposition of DLC films on the AISI 316L stainless steel substrate and their structural properties were investigated under different substrate bias voltages by using molecular dynamics simulations. The results showed that the presence of the $sp^3$ fraction in the DLC films increased when the substrate bias voltage was increased from 0 V to 120 V. The highest magnitude of $sp^3$ fraction (48.5%) was found out at the 120 V bias voltage [4].

In the previous works of our research team [18–20], coatings based on carbon nitrides were deposited by electron-beam physical vapor deposition. The presence of $sp^2$ and $sp^3$ bonds was observed in the formed films. The quantity of C–N bonds increased with the substrate temperature. Nano-clusters consisting of $\beta$–$C_3N_4$ bonds were detected and related to carbon with $sp^2$ coordination.

It is obvious that both techniques for surface modification, namely electron-beam treatment and the formation of coatings, are viable methods for improving the properties of the materials. However, the combination of both methods, including electron-beam surface

modification, and deposition of DLC films has not been studied yet. As already mentioned, the electron-beam surface modification procedure leads to the formation of a significantly finer microstructure and improved mechanical properties [9,10]. In this case, the problem related to the large difference in the hardness between the substrate and subsequently deposited coating is overcome, leading to an improvement in the adhesion and functional properties of the deposited film [21].

In the present study, we investigated the possibility for surface modification of 304-L stainless steel by electron-beam treatment and followed deposition of DLC film at different technological conditions. This development is expected to provide knowledge for a better understanding of the influence of electron-beam treatment on the properties of the obtained films and their application in biomedicine.

## 2. Materials and Methods

304-L SS with the chemical composition 0.029% C, 0.3% Si, 1.6% Mn, 0.026% P, 0.001% S, 0.065% N, 18.06% Cr; 8.0% Ni in wt.%, with a diameter of 20 mm and thickness of 4 mm was used as a substrate material in the present work. Before the deposition of the coatings, the surface of the samples was modified using an electron beam. The experiments were carried out using Leybold Heraeus electron-beam equipment. During the electron-beam treatment process, the working pressure was $2 \times 10^{-4}$ Pa; the accelerating voltage was 50 kV; the electron beam current was 20 mA and 30 mA, corresponding to a beam power of 1000 W and 1500 W; the speed of the specimen motion was 20 mm/s; the electron beam scanning frequency was 1 kHz. It should be noted that the experiments were realized using a linear manner of scanning. During the experiments, the specimen moves horizontally with a constant velocity in a perpendicular direction of the scanning of the electron beam. The treatment process took place for one second. In this case, the beam trajectory does not overlap, leading to an increase in the cooling rate. The scheme of the electron-beam setup for surface modification is presented in Figure 1.

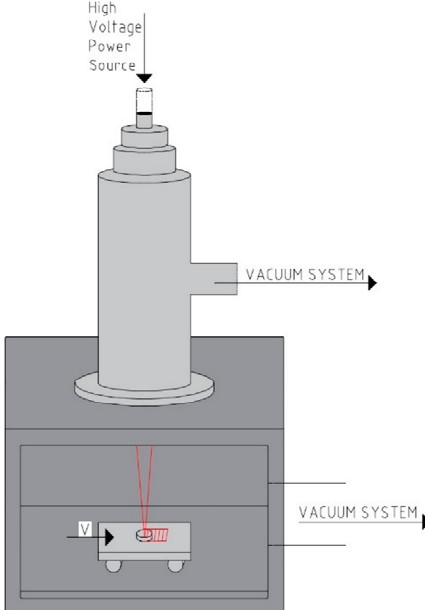

**Figure 1.** A scheme of electron-beam equipment for surface modification.

DLC layers were deposited on already-modified surfaces of the substrate material by electron-beam physical vapor deposition. During the deposition of the coatings, the accelerating voltage was 50 kV, the beam current was 17 mA, the focusing current was 460 mA, the deposition time was 90 s, and the substrates were preheated to 500 °C, corresponding to a thickness of about 1 μm in all cases. For comparison, DLC film was fabricated on an untreated substrate.

The vibrational properties of DLC films were analyzed by Fourier transform infrared (FTIR) spectroscopy. The experiments were carried out with Shimadzu FTIR Spectrophotometer IRPrestige-21 in the spectral range 350–4000 cm$^{-1}$. The spectra were taken in reflectance mode using specular reflectance attachment SRM-8000. Al mirror or the corresponding bare substrate was used as background. The absorption spectrum can be calculated from the measured reflection spectrum by the Kramers-Kronig transformation.

Jobin Yvon Labram spectrometer with a CCD detector was used for the Raman measurements. The wavelength of the laser radiation was 632.8 nm, where the absolute accuracy was about 1 cm$^{-1}$. The laser beam was focused on a spot with a diameter of about 1–2 µm on the surface of the sample by microscope optics. The Raman spectra were measured from 400 to 2500 cm$^{-1}$.

An Atomic Force Microscope (AFM) MFP-3D, Asylum Research, Oxford Instruments with a silicon tip Si-AC160TS-R3 with a 10 nm curvature and an elasticity coefficient of k = 26 N/m was used to scan an area of 400 µm$^2$. Final data regarding the nanoroughness of the samples, along with a 3D image of the scanned area was obtained using the Igor Pro 9 software.

The corrosion resistance of the samples was probed as described previously [22]. In brief: the corrosion potential was measured in phosphate-buffered saline, containing 0.9% potassium chloride under equilibrium conditions, i.e., at zero current flowing through the electrochemical cell. The ability of the DLC coatings to protect the steel surface from oxidation has been judged by the impedance spectra, acquired over the frequency range from 100 kHz to 1 Hz with 15 frequencies per decade (electrochemical impedance spectroscopy, EIS). Measurements were performed in a 0.1 M KCl aqueous solution containing 5 mM ferri/ferro hexacyano ferrates. Polarization characteristics of the samples were determined under conditions, mimicking the physiological ones, in 0.1 M phosphate buffer with pH = 7.4 and were recorded over the potential region from −0.2 V to 0.8 V vs. Ag/AgCl, sat. KCl reference system. Before measurements, each sample was connected to a Pt wire, then the contact and the sides of each sample that have not been coated with DLC layer were sealed with an insulating layer of epoxy resins. For corrosion studies, a 1 cm$^2$ geometric surface area of DLC coated sample was chosen, the rest was sealed with an insulating coating (epoxy resin). Before sealing, a Pt wire was attached to the sample, so that the electrical characteristics can be measured. The electrochemical measurements have been performed in a single compartment electrochemical cell with a working volume of 20 mL in a three-electrode configuration. The samples were connected as working, an Ag/AgCl, sat. KCl was used as a reference, and a Pt foil as an auxiliary electrode, respectively. All electrochemical measurements have been done with a potentiostat-galvanostat Autolab 302 N (Metrohm, Switzerland). The solutions were prepared with ultrapure water (Adrona B30 Bio, Adrona, Lithuania) and ACS reagent grade chemicals.

## 3. Results and Discussion

FTIR spectroscopy is a method for the determination of functional groups and indicates different vibrational modes of various bonds which are present in the films. Figure 2 shows the FTIR spectra of the DLC film deposited on an untreated substrate and electron-beam-treated material in the range of 400–4000 cm$^{-1}$ as a function of electron beam power. For all DLC films, it was observed a clear peak at 672 cm$^{-1}$ corresponding to symmetrical valence oscillations of carbon–carbon bonds [23], and according to the other authors [24], this is a torsional vibration originating from the C–O bond. At the low wavenumber region, there is one characteristic peak for all DLC films at 1515 cm$^{-1}$, which can be assigned to sp$^2$ C vibration mode [25] or attributed to the mixed C–C bonds in sp$^2$ and sp$^3$ hybridization, which evidences the presence of graphite-like and diamond-like phases in the obtained DLC films [26]. On the other hand, this infrared absorption can be assigned to CC and CN triple bond stretching [27]. The peak positioned at a wavelength of 1540 cm$^{-1}$ is related to C=N bonds [28] or graphite-like sp$^2$ bonded carbon [29], but other researchers [26] have reported that this band corresponds to mixed sp$^2$-sp$^3$ C–C bond vibrations.

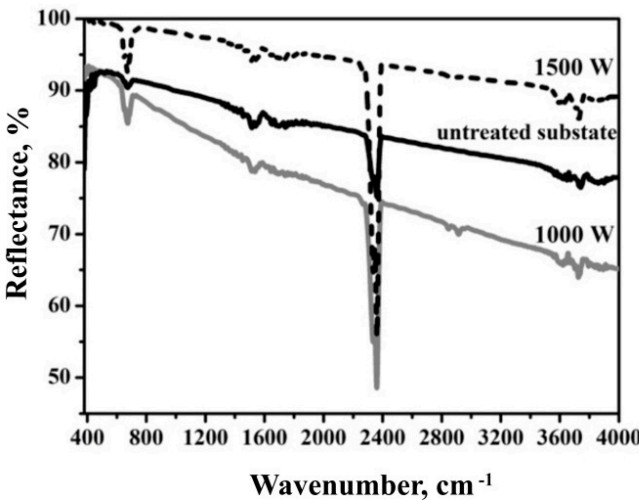

**Figure 2.** FTIR spectra of DLC films deposited on untreated and electron beam treated substrates.

Besides, the peak at 1540 cm$^{-1}$ can be identified as C=C stretching in aromatic sp$^2$ mode [30]. Although this band should not be infrared active, its appearance is probably due to polarization from nearby oxygen atoms. Embedded in the broad absorption background induced by C≡N stretching vibration [30] or environmental CO$_2$ [31], a strong absorption band due to stretching was found at a wavelength of 2362 cm$^{-1}$ attributed to C=N bonds. At the high wavenumber region (Figure 3), four peaks appeared at wavelengths of 2750 cm$^{-1}$, 2848 cm$^{-1}$, 2917 cm$^{-1}$, and 2953 cm$^{-1}$, respectively. The band at 2750 cm$^{-1}$ indicates the presence of ethyl group [32] and the other peaks correspond to sp$^3$–CH$_3$ configuration [33], sp$^3$–CH$_2$ (asymmetrical) [34], and sp$^3$–CH$_3$ (asymmetrical) bonds [35]. The asymmetrical vibration sp$^3$–CH$_2$ is due to the sp$^3$-hybridized carbon to hydrogen and tetrahedral carbon structure [36], and is the highest absorption band at 1000 W. Additionally, it has been reported as a result of hydrogen bonds in the form of sp$^3$–CH$_2$ and sp$^3$ CH$_3$ groups [37]. The broad absorption between 3600 cm$^{-1}$ and 3800 cm$^{-1}$ might be contributed by free OH bonds [38] and O–H stretching vibrations of unassociated OH groups [30,31] owing to the presence of a small amount of carboxylic acids. Based on the performed analysis, it can be concluded that the peaks around 1515 cm$^{-1}$, 1540 cm$^{-1}$, 2848 cm$^{-1}$, 917 cm$^{-1}$, and 2953 cm$^{-1}$ represent the vibration of the sp$^2$ and sp$^3$ C–C bonds and indicate a presence of graphite and diamond-like phases in the deposited films.

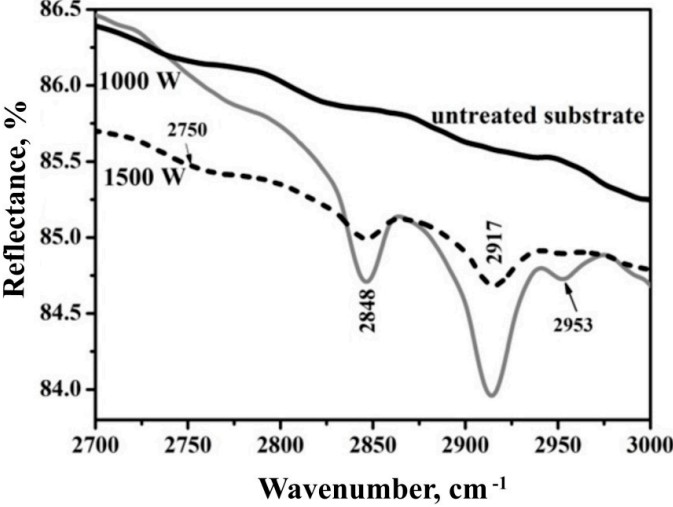

**Figure 3.** FTIR spectra of DLC films obtained on untreated substrates and electron-beam-treated substrates in the spectral range of 2700–3000 cm$^{-1}$.

The experimentally obtained Raman spectra of the DLC films deposited on untreated and electron-beam-treated 304-L stainless steel substrates are shown in Figure 4. The results reveal the existence of D and G bands that are typical for DLC-based coatings and are located at about 1350 cm$^{-1}$ and 1560 cm$^{-1}$, respectively [39]. The existence of the G peak is attributed to the C–C bonds, while the breathing mode of carbon rings is the reason for the appearance of the D band [34]. Therefore, the bond structure within the DLC coating can be studied by a characterization of the above-discussed bands. The precise determination of the G-band position and the full width at a half maximum (FWHM) is directly related to the carbon bonding structure. The increase in the peak position of the G-band corresponds to an increase in the sp$^2$/sp$^3$ ratio, while higher values of the FWHM are associated with a lower number and dimension of sp$^2$ clusters [40]. The experimentally obtained data for the peak position and FWHM of the G-band are summarized in Table 1.

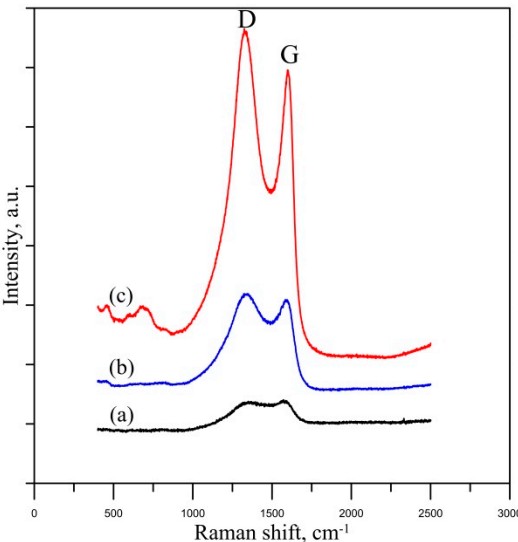

**Figure 4.** Raman spectra of deposited DLC coatings on: (**a**) untreated; (**b**) electron-beam-treated substrates at 1000 W, and (**c**) electron-beam-treated substrates at 1500 W.

**Table 1.** Peak position and FWHM of the G-band.

| Parameters of the G-Band- | Untreated Substrate | 1000 W | 1500 W |
|---|---|---|---|
| Peak position, cm$^{-1}$ | 1578 | 1596 | 1587 |
| FWHM, cm$^{-1}$ | 167 | 93 | 106 |

Considering the peak position of the G-band of the coating deposited on an untreated substrate, it is about 1578 cm$^{-1}$, and it increases to 1596 cm$^{-1}$ and 1587 cm$^{-1}$ in the case of deposition of DLC on electron-beam-treated substrate by a beam power of 1000 W and 1500 W, respectively. This means that the preliminary electron-beam modification of the 304-L stainless steel substrate leads to an increase in the sp$^2$/sp$^3$ ratio, where in the case of EBT with lower beam power, it is higher in comparison with the treatment by 1500 W. This means that the amount of diamond-like structure (i.e., sp$^3$ bonds) is lower in the case of the substrate treatment by a higher power. However, the peak position of the G-band has the lowest value at the coating deposited on untreated substrate, meaning that the discussed ratio is the lowest, i.e., the amount sp$^3$ is the highest. Nevertheless, the difference in the position is in the range of 10 cm$^{-1}$, meaning that the amount of the sp$^3$ is very similar in all considered cases.

Similar conclusions can be drawn by considering the FWHM of the G-band. It was found that the width of the peak in the case of an untreated substrate is 167 cm$^{-1}$. After

the application of the electron-beam treatment procedure, the FWHM decreases to 93 cm$^{-1}$, at a beam power of 1000 W, and 106 cm$^{-1}$ at 1500 W. This means that in the case of an untreated substrate, the number of the clusters of sp$^2$ bonds and their size is much smaller in comparison with the DLC films deposited on previously treated one. It should be noted that the change in the number and dimension of sp$^2$ clusters does not always lead to transformation in the sp$^2$/sp$^3$ ratio and some sp$^2$ bonds would not form clusters.

It can be concluded that the DLC coating deposited on an electron-beam-treated substrate by a beam power of 1500 W consists of a higher amount of sp$^3$, followed by a small decrease in the case of a substrate treatment by a beam power of 1000 W. The DLC coating deposited on untreated substrate represents the highest concentration of sp$^3$ bonds. However, the shift in the peak position of the G-band is insignificant, meaning that the relative contribution of the sp$^3$ bonds in the deposited DLC coatings is relatively unchanged in all cases.

The results obtained by FTIR and Raman experiments confirm the presence of DLC coatings deposited on the steel substrate. As already mentioned, according to the definition, DLC coatings [14] are hard and amorphous-like structures with a mixture of sp$^2$/sp$^3$ and hydrogen, where the amount of sp$^3$ bonds is significant. It was found that the structure of the coatings deposited on treated and untreated stainless-steel substrates exhibits a mixture of sp$^2$/sp$^3$ bonds, as well as CH$_2$ groups in the sp$^3$ hybridization.

The surface topography of the considered DLC films is shown in Figure 5. The films were deposited on previously polished and electron-beam-treated 304-L stainless steel substrates by a beam power of 1000 and 1500 W, at a temperature of 500 °C by electron beam physical vapor deposition. For comparison, DLC film was fabricated on an untreated substrate to follow the influence of the material surface modification on the nano-roughness of the as-deposited films. It was found that the surface nano-roughness of the coating on an untreated substrate is only 15 nm. The surface roughness of the DLC films on treated substrates was 101 nm for the treated substrate by a beam power of 1000 W, and 48 nm at a beam power of 1500 W, respectively. Therefore, with an increase in the beam power, the surface roughness decreases in the present particular case. Thus, it can be concluded that the electron-beam treatment of the substrate leads to an increase in surface roughness.

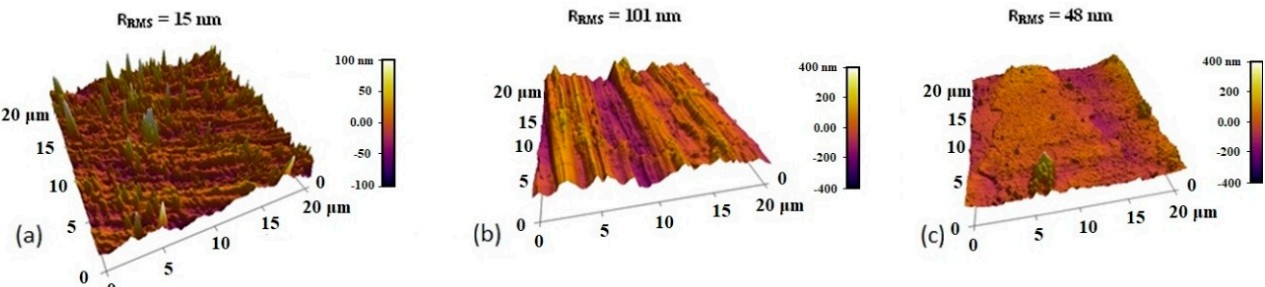

**Figure 5.** Three-dimensional AFM micrographs of the surface topography of the deposited DLC coatings on (**a**) untreated substrate; (**b**) treated substrate with a beam power of 1000 W; (**c**) treated substrate with a beam power of 1500 W.

According to the literature, the application of treatment with an electron beam has a different influence on the resultant surface roughness [41]. In the case of treatment of flat surfaces, during the EBT process, some amount of the treated material is evaporated and subsequently condensed, which is capable to form protrusions, and increasing the roughness. On the other hand, the electron-beam treatment procedure of rough surfaces leads to melting of the peaks. The molten material flows to the valleys and increases in smoothness [41]. These statements are in agreement with the results of our study, where the EBT of the base material leads to an increase in its roughness, and therefore, to the coatings since they follow the topography of the substrate.

The results obtained show that in both cases of electron-beam treatment of the substrate leads to an increase in the roughness, where this effect is more pronounced in the case of a lower value of the beam power of 1000 W. In this case, the initial surface roughness of the substrate was replaced by a wave-like topography. The electron-beam treatment procedure leads to melting the surface, where fluid flows, acting from the surface to the depth of the molten material, are formed due to the high-temperature gradient and are responsible for the formation of the observed surface topography. With an increase in the beam power to 1500 W, it is obvious that the surface of the specimen is significantly smoother, which could be attributed to the significantly longer lifetime of the molten material.

It should be noted that these results have significant importance and can be directly implemented in modern biomedicine and implant manufacturing. Higher surface roughness corresponds to a larger contact surface area. This is accepted to be of major importance for cell adhesion, proteins adsorption, biomineralization phenomenon, etc. [42]. The authors of [43] showed that the increased roughness of 13 nm leads to a higher amount of attached cells and cell density after 16 h of incubation in comparison with the 5 nm rough surface.

Corrosion potentials of the samples were determined at open circuit, i.e., at no current flowing through the cell, until an equilibrium potential was established. The authentic records of the samples' potentials vs. time are depicted in Figure 6. As can be seen, the equilibrium potentials of the three specimens have positive values that increase with the rise of the beam power from 1000 to 1500 W, with the one of the thermally pretreated sample (untreated sample) laying above them, most probably due to structural surface changes resulting from the thermal treatment during the deposition, i.e., the differences between them are not so substantial, but the 1000 W sample has the lowest one. The potential range is exactly 1 V, starting at $-0.2$ V and reaching an upper limit of $+0.8$ V. Usually, the tendency of iron alloys to corrode is electrochemically detectable within this region. It should be noted that all the potentials are given vs. Ag | AgCl, KCl sat. reference system, the potential of which is $+0.199$ V vs. normal hydrogen electrode (i.e the studied potential range is from $-0.001$ to $+0.999$ V vs. NHE). Depending on the pH of the background electrolyte, at slightly higher applied potentials, electrolysis of the electrolyte solution would start. The corrosion potential of the DLC-coated samples is 0.5–0.6 V more positive than the corrosion potential of bare stainless steel ($\sim-0.4$ V vs Ag | AgCl, sat. KCl), which means that the bare steel sample will start to dissolve spontaneously in an aqueous environment, while the DLC-coated samples would not corrode under equivalent conditions. The measured corrosion potentials suggest that these samples are corrosion-resistant even in a medium, containing corrosive agents such as chloride ions.

These findings were confirmed by the EIS studies, shown in Figure 7. The impedance spectra of the samples indicate the formation of a homogeneous protective layer over the metallic surface, as it can be deduced from the large semicircle observed for DLC film obtained at 1500 W. EIS is an alternating current electrochemical technique that is used to study the sample's behavior at the surface-solution interface in the presence of redox species. The EIS spectra were subjected to simulations, aiming at revealing the reasons for exhibited impedance behavior. The modeling gave analogous equivalent electrical circuits for 1000W and 1500W specimens (semicircle corresponding to charge transfer resistance, without diffusion region). For the untreated sample, the simulation of the EIS spectrum resulted in a circuit consisting of some very large number of constant-phase elements: a similar picture is usually observed for electrode surfaces with nanoformations (which are well visible on the AFM surface topography image) [44]. The EIS spectra represent either a straight line with a slope greater than 45° (a slope of 45° is indicative for the diffusional region, known also as Warburg impedance) or a slightly curved line, albeit without a clearly expressed semicircle. This means that we cannot define Rct for the untreated sample so that to compare it with the corresponding values of the other two samples. The appearance of semicircles points toward the inability of redox species to reach the conductive surface so as to reduce or oxidize over it, and the semicircle's diameter increases with the density of the protective layer. Thus, the diameters of the semicircles of the impedance spectra of

the samples under study sharply increase as the intensity of the beam power is increased from 1000 W to 1500 W, indicating that the firmness of the protective diamond-like carbon layer increases. A much smaller charge transfer resistance of the DLC layer at 1000 W is indicative of unprotected zones on the metal surface.

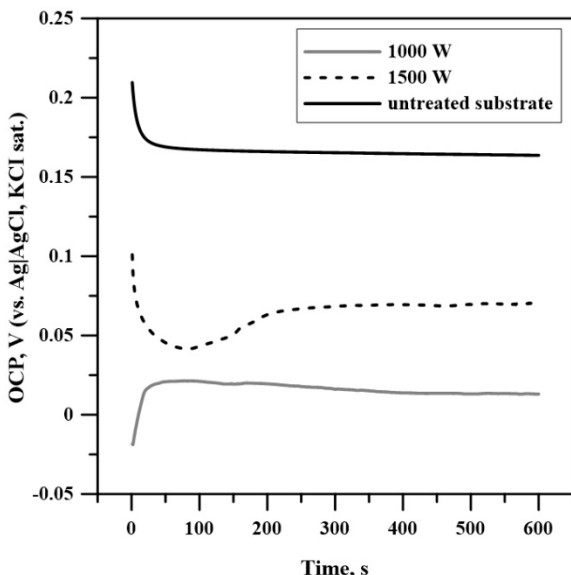

**Figure 6.** Corrosion potential of DLC-coated steel in 0.9% KCl solution as a function of time for reaching equilibrium; electrolyte: phosphate buffered saline, pH = 7.4 containing 0.9% KCl.

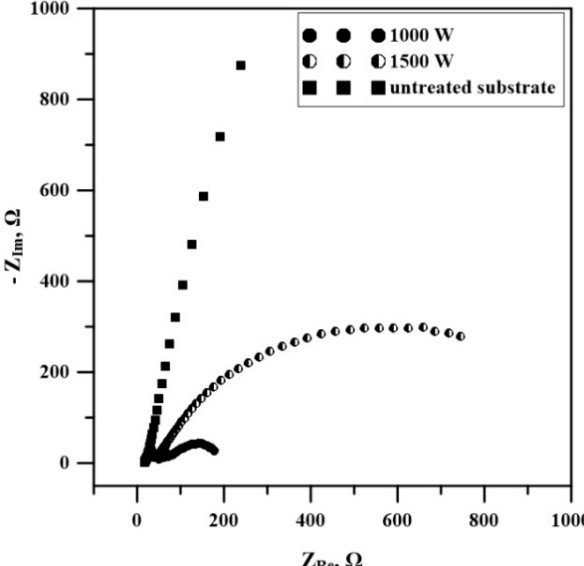

**Figure 7.** Impedance spectra of DLC-coated steel samples, intensity of the laser beam is indicated close to each spectrum; electrolyte: 0.1 M KCl, containing 5 mM $K_4[Fe(CN)_6]$ and 5 mM $K_3[Fe(CN)_6]$.

In support of this finding, the cyclic voltammograms of the sample at 1000 W reveal a broad peak at 0.2 V that might result from the oxidation of an unprotected metal surface (Figure 8). Cyclic voltammetry is a direct current potentiodynamic electrochemical technique that records the current value as a function of the applied potential and is usually applied to diagnose the existence of oxidative or reductive processes at the electrode-solution interface. Since no redox-active species were present during the voltammetric studies, performed in phosphate buffer with pH = 7.4, the peak is obviously due to the corrosion of the metal surface, i.e., to the oxidation of iron. The DLC coating deposited on an electron-beam-treated substrate by a beam power of 1000 W is not continuous due

to, most probably, the observed picture caused by a tiny crack (mechanical injury) on the studied surface, which gives the hump appearing in Figure 8 around the potential of Fe dissolution; however only if the sample is under polarization. If the surface was coated without breaks, the hump would not appear, and the Rct would appear much greater since electrochemical studies give an overall surface characterization. According to the authors of [45,46], the surface topography of the film deposited on rough substrates is attributed to the growth defects, caused mostly by the roughness of the base material. The existence of these growth defects can be considered as the main reason for the deterioration in the corrosion properties of the considered specimen (i.e., the DLC film deposited on an electron-beam modified substrate by a beam power of 1000 W). The other two samples behave as inert conductive surfaces that are not prone to corrosion. The DLC film on the untreated substrate shows a slight tendency to dissolution at polarization potentials exceeding 0.5 V that is due to lack of a protective coating. The DLC film deposited on an electron-beam-modified substrate with a beam power of 1500 W shows no signs of destruction despite the large current passing through the interface. As mentioned above, the preliminary electron-beam modification leads to the formation of a significantly finer microstructure and rise in the hardness [9,10], where this effect is much more pronounced in the case where the cooling rate is higher, i.e., at a higher beam power. This means that the treatment of the substrate by the higher beam power is capable to overcome the problem related to the difference in the hardness between the film and the substrate improving the adhesion of the coating. Furthermore, the treatment procedure by a beam power of 1500 W leads to the formation of a smoother surface than that obtained by 1000 W, meaning that the amount of growth defects, which are responsible for the deterioration of the corrosion properties, will be lower. These statements are consistent with the results obtained in the present study, where a DLC coating deposited on an electron-beam-treated substrate with a beam power of 1500 W did not exhibit signs of destruction. The slight tendency of dissolution of the film deposited on an untreated substrate could be attributed to the significant mismatch of the hardness between the substrate and coating. In this case, the adhesion is deteriorated, leading to worsening of the corrosion resistance.

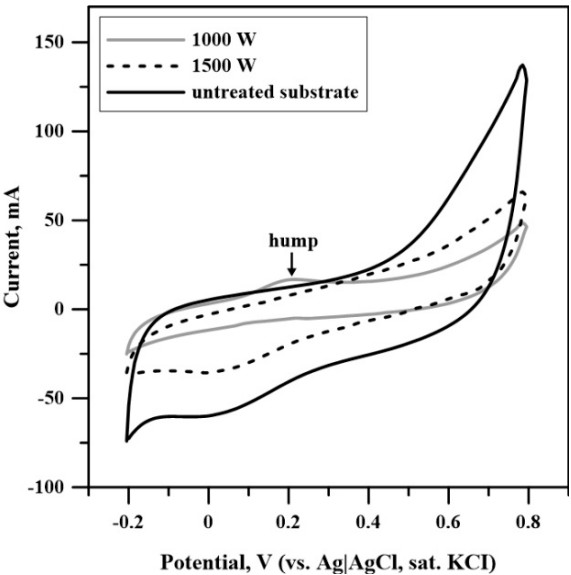

**Figure 8.** Polarization curves (cyclic voltammograms) of DLC-coated steel samples, scan rate 0.1 V/s; electrolyte phosphate buffer, pH = 7.4, room temperature.

The results obtained in the present study show the possibility for modification of the surface properties of 304-L stainless steel by a duplex surface modification approach including electron-beam treatment and subsequent deposition of the diamond-like coating. It was found that the preliminary modification of the substrate material by electron beam

leads to an increase in surface roughness. Generally, this could have some benefits from a practical point of view. Higher values of the surface roughness correspond to a larger amount of the contact surface area. This could be appropriate for cell growth and adhesion, proteins adsorption, biomineralization phenomenon, etc., which is of major importance for implant manufacturing and implementation of the materials in modern biomedicine. On the other hand, higher surface roughness corresponds to deterioration in the corrosion resistance, which is one of the main problems for modern biomedical materials. In this case, separations of metallic ions occur, which leads to implant failure and disadvantageous reactions [47]. Therefore, the technological conditions of the treatment technologies have to be optimized very precisely in order to form a surface with sufficient roughness for cell growth and adhesion support and at the same time, appropriate corrosion properties. In the present study, we demonstrated that the electron-beam surface treatment of the 304-L stainless steel substrate by a beam power of 1000 W leads to a significant increase in the surface roughness (more than six times), but also to deterioration in the corrosion properties of the deposited DLC film. Higher beam power of 1500 W leads to the formation of a smoother surface than that of the specimen subjected to an electron-beam treatment by a beam power of 1000 W (to about three times) and improvement in the corrosion resistance. In this case, the adhesion between the coating and the substrate is greatly improved, leading to an enhancement in the corrosion properties. The specimens are studied with a further aim to be tested as suitable materials for implants, where many other specifics such as protein adsorption, biocorrosion, etc., will be investigated.

## 4. Conclusions

In the present study, we present the possibility of modifying the surface topography and corrosion properties of 304-L stainless steel through the application of duplex surface modification approach by electron-beam treatment and subsequent deposition of the diamond-like carbon (DLC) coatings. The results obtained showed that the preliminary electron-beam treatment procedure does not significantly influence the amount of $sp^3$ bonds, as well as on the $sp^2/sp^3$ ratio at the deposited DLC coatings. The application of the electron-beam treatment procedure leads to an increase in the surface roughness in both cases, where the highest values were measured at the specimen treated by a beam power of 1000 W. The higher nano-roughness leads to an increase in the contact surface area, which is of major importance and has a significant advantage in the support of cell growth and adhesion. The investigated corrosion properties exhibit that the samples considered in the present study are corrosion resistant even in a medium containing corrosive agents such as chloride ions. It can be concluded that the most corrosion-resistant specimen is DLC coating deposited on electron-beam-treated 304-L SS substrate by a beam power of 1500 W.

**Author Contributions:** Conceptualization, S.R. and S.V.; methodology, S.R., G.K., M.S., P.R., M.O., V.S., N.D. and S.V.; formal analysis, S.R., G.K., M.S., P.R., M.O., V.S., N.D. and S.V.; investigation, S.R., G.K., M.S., P.R., M.O., V.S., N.D. and S.V.; writing—original draft preparation, S.R. and S.V.; writing—review and editing, S.V.; project administration, S.R. All authors have read and agreed to the published version of the manuscript.

**Funding:** This research was funded by the Bulgarian National Science Fund, grant number KP-06-M-37/1.

**Institutional Review Board Statement:** Not applicable.

**Informed Consent Statement:** Not applicable.

**Data Availability Statement:** Not applicable.

**Acknowledgments:** Electrochemical studies were performed thanks to the research infrastructure of the Center for Competence "Personalized Innovative Medicine, PERIMED (BG Programme "Science and Education for Smart Growth" grant BG05M2OP001-1.002-0005-C01)".

**Conflicts of Interest:** The authors declare no conflict of interest.

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
