# Peer review of "Duplex Surface Modification of 304-L SS Substrates by an Electron-Beam Treatment and Subsequent Deposition of Diamond-like Carbon Coatings"

_coatings, doi:10.3390/coatings12030401_

Round 1

Reviewer 1 Report

The publication concerns the modification of 304-L SS steel. The work is written in a comprehensible way and may be interesting for the audience, although there are several problematic aspects listed below.

  1. The title suggests that a diamond-like layer has been deposited on the steel. However, the presence of diamond cannot be seen from either FTIR or Raman studies.
  2. There is no information about the electrode surface or how it was determined
  3. I would ask the authors to draw the sp3 hybridization of CH2 group
  4. Why is the corrosion potential higher for the 1000 W sample, and the EIS shows that the 1500 W sample has the highest Rct value?
  5. Why is the potential window so narrow? For diamond layers, the window width should be much greater.
  6. The authors write that the 1000W sample was not continuous, hence the peak on the CV appears. Does this mean that the electrochemical measurements were made only for 1 sample? What would be the effect if the 1000W sample was continuous?
  7. The introduction lacks other examples of carbon materials covering steel, e.g. doi: 10.3390/coatings11111403

Author Response

Dear Reviewer,

The authors would like to thank you for the valuable comments. They are very important for us.  Attached you can find our responses. 

Reviewer 2 Report

The electrochemical measurements have been performed in a single compartment electrochemical cell with working volume of 20 mL in a three-electrode configuration. The samples were connected as working, a Ag/AgCl, sat. KCl was used as a reference, and a Pt foil – as auxiliary electrode, respectively. All electrochemical measurements have been done with a potentiostat-galvanostat Autolab 302N (Metrohm, Switzerland).

In order to minimize its impact on the potential measurement, a L-type Lukin capillary Sand Core Salt Bridge Reference Electrode Outer Salt Bridge should be used. The Paper is excellent.

Author Response

The Authors greatly appreciate Reviewer’s high esteem and their valuable recommendation to work with a separated-compartments electrochemical cell with Lugin capillary, which is not currently available in the Laboratory where corrosion studies have been done. We will try to borrow/order such a special cell for our future studies.

Reviewer 3 Report

The authors may need to provide TEM or HRTEM to verify the phase structure of the coating.

The paper was well written, I am afraid the findings reported in the paper are not so excited, the corrosion resistance is not improved apparently, and the findings of diamond like carbon were not verified by providing atomic level structures, e.g. high resolution transmission electronic microscopes.

Author Response

Dear Reviewer,

The authors highly appreciate your comments. 

 We completely understand that the verification of the DLC coatings by high-resolution transmission electron microscopy will result in significant improvement of this part of the manuscript. However, we cannot obtain such results during the time for the revision. We will pay attention to these aspects for our future works.

The corrosion potential of the DLC-coated samples is ca. 0.5 – 0.6 V more positive than the corrosion potential of bare stainless steel (~ - 0.4 V vs Ag|AgCl, sat. KCl), which means that bare steel sample will start to dissolve spontaneously in an aqueous environment, whilst the DLC- coated samples would not corrode under equivalent conditions. If this reviewer is comparing the equilibrium potentials of the control sample with no electron beam pretreatment and concludes that the corrosion resistance is not improved, then it must be taken into account that the specimens are studied with a further aim to be tested as suitable materials for implants, where many other specifics as protein adsorption, biocorrosion, etc. will be investigated, however, taking into account these studies.

The authors would like to thank you for the valuable comments and suggestions!

Reviewer 4 Report

In this article, the authors introduced e-beam treatment on stainless steel before depositing DLC film. The effect of varying e-beam power on the surface morphology was studied, which was also affecting corrosion resistance property. I have a few remarks:

  1. In Section 2 Materials and Methods, I suggest the authors give more details when describing their experiment design. For example, in Line 101 to 102, the authors did not describe how exactly the specimen is moving in related to the beam scanning, only telling readers the “speed of the motion”. In Fig. 1, it appears that the sample moves horizontally in one direction (x-direction) while the beam scans horizontally in the perpendicular direction (y-direction). If this is correct, I suggest the authors to specify this in the article.
  2. Following the first question, I suggest the authors add the following details. What’s the model of the e-beam etching instrument? What is the vacuum condition? How long did it take to fully etch one sample? How big is the sample size?
  3. In Table 1, the G peak on untreated substrate is at 1578 cm-1, while on 1500 W beam treated substrate it is 1587 cm-1. However, these numbers do not match with the description in the article (Line 205). What are the correct values?
  4. When describing the Raman results, some of the sentences were written in a way that could leave readers very confused. In Line 196 to 197, the authors stated that the increase in the peak position corresponds to an increase in the sp2/sp3 In Line 209 to 210, the authors wrote that “where in the case of EBT with lower beam power, it is lower in comparison with the treatment by 1500 W”. However, what exactly is lower here was not specified. If what the authors implied is that the sp2/sp3 ratio is lower, then it does not match with the peak shift given in the table. The sample pre-treated with 1000 W e-beam has higher G peak shift compared to 1500 W e-beam pre-treated one. So, it should have a higher sp2/sp3 ratio. Or did I miss anything?
  5. In Line 236 to 238, the description of the surface roughness does not match with the description of Fig. 5. I assume that the sample pre-treated with 1000 W e-beam is the roughest, then the sentence in Line 236 to 238 needs to be corrected.
  6. Please rewrite the sentence in Line 273 to 275 in a way to specify that the sample with no pre-treatment has the highest potential, followed by two samples with substrate pre-treatment.
  7. In Line 291 to 292, the authors describe the impedance spectrum of the DLC film obtained on untreated substrate is typical for rough conductive surfaces, without elaborating further details. It caused significant confusion as the DLC film deposited on untreated substrate is in fact the smoothest one according to the AFM results. Also, in Fig. 7, the Nyquist plot of the DLC film with no substrate pre-treatment is part of a semi-circle with a very large diameter, much larger than the other two samples. It indicated that the corrosion resistance is the highest among the three, which does not match the conclusion given by the authors. Is there any specific reason that the authors consider the sample DLC 1500 W has better corrosion resistance?

Author Response

Dear reviewer,

Thank you for your valuable comments and suggestions. They are very important for us. Attached you can find our responses.

Round 2

Reviewer 1 Report

I agree to the publication of the revised manuscript

Author Response

The authors would like to thank the reviewer for the comment and recommendation!

Reviewer 3 Report

The paper has been greatly improved, I noticed a related reference has been missed: Y. Zhang*, X. Yan, W. Liao, K. Zhao, Effects of Nitrogen Content on the Structure and Mechanical Properties of (Al0.5CrFeNiTi0.25)Nx High-Entropy Films by Reactive Sputtering. Entropy. 2018, 20(9):624.

Author Response

The authors would like to thank the reviewer for the comment. The missed reference has been added.

Reviewer 4 Report

After going through the revised manuscript, I have a few remarks,

  1. The revised manuscript seems a bit messy. Example of which is Line 307. The sentence starting with “respected reviewer” seems not appropriate in a manuscript. Another example is the sentence in Line 408 to Line 411. Is it finished? I could not catch what the authors mean here. I think very substantial edit is still needed.
  2. Starting from Line 331, the authors claimed that “a similar picture is usually observed for electrode surfaces with nanoformations”. Is there any reference supporting this claim? If so, please provide this information.
  3. Since the authors claimed that it is difficult to define Rct for untreated sample. I suggest the authors revise the conclusion they made in Line 360 to Line 363. It is now more apparent to me that the untreated sample, with a surface roughness of only 15 nm, has the best corrosion resistance. Especially in Line 397 to 399, the authors claimed that “On the other hand, higher surface roughness corresponds to deterioration in the corrosion resistance, which is one of the main problems for modern biomedical materials”. The authors need to make their conclusions consistent with each other.

Author Response

Dear reviwer!

The authors would like to thank you for rewiewing our paper and showing us some mistakes within the manuscript, made due to the limited time for the revision. We highly appreciate your effort. Attached you can find a list with the answers to your comments and remarks. 

highly appreciated your effort in reviweing our manuscript, and finding some mistakes, mentioned in your reviwe report. 
